# Improvement of Growth and Morphology of Vegetable Seedlings with Supplemental Far-Red Enriched LED Lights in a Plant Factory

**Hyunseung Hwang** [1],[†] , **Sewoong An** [2],[†] , **Byungkwan Lee** [1] and **Changhoo Chun** [1],[3],[*]

[1]  Department of Plant Science, Seoul National University, Seoul 08826, Korea; behong47@snu.ac.kr (H.H.); byukl1301@snu.ac.kr (B.L.)

[2]  Department of Horticultural Crop Research, National Institute of Horticultural and Herbal Sciences, Rural Development Administration, Wanju 55365, Korea; woong0911@korea.kr

[3]  Research Institute of Agriculture and Life Sciences, Seoul National University, Seoul 08826, Korea

[*]  Correspondence: changhoo@snu.kr; Tel.: +82-2-880-4567

[†]  These authors contribute equally to this work.

**Abstract:** Although light-emitting diode (LED) lamps have been broadly applied in horticultural production to improve plant yield and quality, compared to natural light there is a disadvantage in the lack of far-red light in the LED spectrum. Far-red light has been studied widely to control plant growth and development. Therefore, this study aimed to find the effect of supplemental far-red-enriched LED lights to control the growth of tomato, red pepper, cucumber, gourd, watermelon and bottle gourd seedlings. The treatments were cool white LED:far-red LED at ratios of 5:0, 5:1, 5:2 and 5:3. The growth of tomato and red pepper seedlings, including hypocotyl length, was correlated to far-red light and light intensity. The phytochrome photostationary state (PSS) value of maximum hypocotyl length by supplemental far-red-enriched light ranged from 0.69 to 0.77 in tomato and red pepper seedlings. Although hypocotyl lengths of cucumber and watermelon were greatly affected by PSS, the PSS value for maximum hypocotyl length was lower than for tomato and red pepper. These results show that manipulating supplemental far-red enrichment can be used to control vegetable seedling growth with some variation among plant species.

**Keywords:** far-red-enriched light; LED; phytochrome photostationary state; hypocotyl length; seedling

## 1. Introduction

Plant responses to light may vary depending on light intensity, photoperiod and light quality. Among light properties, control of the light spectrum from blue to red has been widely researched for effects on plant growth and development such as enhancing biomass and adjusting morphology [1,2]. In particular, supplemental far-red light has been effectively used to control plant growth and morphogenesis [3]. As far-red light intensity increased, plant growth and morphology were affected in lettuce (*Lactuca sativa* L.) [1,4–6], tomato (*Solanum lycopersicum* L.) [7], squash (*Cucurbita* spp.) [8], red pepper (*Capsicum annuum* L.) [9] and snapdragon (*Antirrhinum majus* L.) [10] seedlings. End-of-day far-red lighting was used as an effective method influencing stem and hypocotyl elongation in watermelon (*Citrullus lanatus*) [11] and tomato rootstocks [7]. Research on red to far-red ratio manipulation showed effects on growth and photomorphogenesis including stem development in many plant species, such as soybean (*Glycine max* (L.) Merr.) [12], cucumber (*Cucumis sativus* L.) [13], bottle gourd (*Lagenaria siceraria*) [14], common bean (*Phaseolus vulgaris* L.) [15] and red pepper [9] seedlings.

In the use of artificial lighting such as fluorescent, metal-halide, high-pressure sodium vapor and light-emitting diode (LED) [16] lamps, light intensity and quality of each show clear differences

from natural light, especially in the far-red range [17]. Plants show a shade avoidance response (SAR) under natural lighting [18] including effects on stem elongation and dry mass partitioning to the shoot, and a reduction of the red to far-red light ratio under artificial lights affects the SAR [3,19]. Phytochromes, as one of the most important photoreceptors, enable plants to recognize the red to far-red ratio by perceiving the light conditions [20,21]. Phytochrome responses under artificial lights are generally classified into four groups by their light characteristics (VLFR: very low fluence response; LFR: low fluence response; R-HIR: red high irradiance response; and FR-HIR: far-red high irradiance response) [20,22]. Among these phytochrome responses, FR-HIR reactions can explain photoinhibition of stem elongation, and these might be induced by prolonged light irradiation with far-red light, although it depends on light intensity [23–26]. The phytochrome photostationary state (PSS), defined as the ratio of active phytochrome (Pfr) to total phytochrome (Pfr+Pr), has been used to quantify stem extension and elongation responses to light intensity and quality. PSS value from 0.70 to 0.85 has shown a negative linear correlation between shoot lengths of the plant [27–30].

Among artificial lights, LEDs have recently become more widely used in horticultural production to more easily control light quality [31]. However, because LED lamps are deficient in far-red light, supplemental far-red-enriched lighting could effectively regulate plant growth and morphology. Therefore, this study aimed to determine the effects of supplemental far-red-enriched light intensities for controlling the growth of young seedlings of tomato, red pepper, cucumber, gourd, watermelon and bottle gourd.

## 2. Materials and Methods

### 2.1. Plant Material and Cultivation Conditions

Seeds of tomato (*Solanum lycopersicum* L. cvs. Dotaerangdia and B-blocking, Takii, Korea), red pepper (*Capsicum annuum* L. cvs. Shinhong and Tantan, Nongwoo Bio, Korea), cucumber (*Cucumis sativus* L. cv. Joeunbaegdadagi, Seminis, Korea), gourd (*Cucurbita ficifolia* Bouché cv. Heukjong, Hungnong Seeds, Korea), watermelon (*Citrullus vulgaris* L. cv. Sambokkul, Hungnong Seeds, Korea) and bottle gourd (*Lagenaria siceraria* (Mol.) Standley cv. Bulrojangsaeng, Syngenta, Korea) were sown into commercial soil mix (Plant World, Nongwoo Bio. Co., Ltd., Yeoju, Korea) in a 128-cell plug tray (2.8 × 2.8-cm; 18-mL volume) and cultivated in a closed system with LED lighting. The seedlings were grown at an air temperature of 26/22 °C (light/dark periods), a photosynthetic photon flux (PPF) of 200 $\mu mol \cdot m^{-2} \cdot s^{-1}$ with a 16/8h light photoperiod, a $CO_2$ concentration of 600 $\mu mol \cdot mol^{-1}$ and a relative humidity of 60%. PPF in all treatments was measured at the top of the tray (28-cm distance from the lamps). Seedlings were subirrigated with Yamazaki [32] nutrient solution (pH 6.5 and EC 0.5–1.5 $dS \cdot m^{-1}$) every 2 or 3 days.

### 2.2. Supplemental Far-Red Lighting Treatments

Light treatments began after germination. Using 10 cool white LEDs (T5/20W6500K, Parlux, Incheon, Korea), supplemental far-red LED (HT251-Far-red, Bissol LED) lights were used at three different intensities with a peak wavelength of 712 nm. The spectral distribution of the light sources was measured with a spectroradiometer (BLUE-Wave spectrometer, StellarNET Inc, Tampa, FL, USA) (Table 1, Figure 1). Cool white LEDs were distributed throughout the photosynthetically active wavelength range (400–700 nm) and characteristically accounted for a greater percentage of green-yellow light (500–600 nm). The treatments were the numbers of cool white LED lights:far-red LED lights at ratios of 5:0, 5:1, 5:2 and 5:3, or W5F0, W5F1, W5F2 and W5F3, respectively.

In each treatment, the far-red-enriched LEDs were on at the same time as the cool white LEDs, creating four different phytochrome photostationary state (PSS) values. The average quantum per 1-nm wavelength of each treatment from 300 to 800 nm was collected at 12 points using a LI-190

quantum sensor (LI-COR Inc., Lincoln, NE). The estimated PSS of each treatment was calculated from the spectral distribution data following the method of Sager [27]:

$$\mathrm{PSS} = \sum_{300}^{800} \mathrm{N}(\lambda)\sigma(\mathrm{r}\lambda) / \left( \sum_{300}^{800} \mathrm{N}(\lambda)\sigma(\mathrm{r}\lambda) + \sum_{300}^{800} \mathrm{N}(\lambda)\sigma(\mathrm{fr}\lambda) \right)$$

where $\mathrm{N}(\lambda)$, $\sigma(\mathrm{r}\lambda)$ and $\sigma(\mathrm{fr}\lambda)$ indicate the photon flux and the photochemical cross-section of Pr and Pfr, respectively. The PSS value decreased as the supplemental far-red-enriched light intensity increased. PSS of W5F0, W5F1, W5F2 and W5F3 treatments were 0.85, 0.69, 0.63 and 0.60, respectively (Table 1).

**Table 1.** Light intensity and phytochrome photostationary state values of supplemental far-red-enriched lights.

| Treatment | Light Intensity (µmol·m$^{-2}$·s$^{-1}$) | | | | PPF [z] | TPF [y] | PSS [x] |
| --- | --- | --- | --- | --- | --- | --- | --- |
| | 400–500 nm | 500–600 nm | 600–700 nm | 700–800 nm | | | |
| W5F0 | 42.4 | 122.6 | 33.2 | 1.6 | 198.2 | 199.8 | 0.85 |
| W5F1 | 46.0 | 146.8 | 58.2 | 18.9 | 251.0 | 269.9 | 0.69 |
| W5F2 | 49.1 | 168.1 | 80.2 | 34.1 | 297.4 | 297.4 | 0.63 |
| W5F3 | 53.2 | 195.3 | 108.3 | 53.4 | 356.8 | 356.8 | 0.60 |

[z] PPF: photosynthetic photon flux (photon flux integral from 400 to 700 nm, in µmol·m$^{-2}$·s$^{-1}$). [y] TPF: total photon flux (photon flux integral from 400 to 800 nm, in µmol·m$^{-2}$·s$^{-1}$). [x] PSS: phytochrome photostationary state, which is an estimated value following Sager et al. [27].

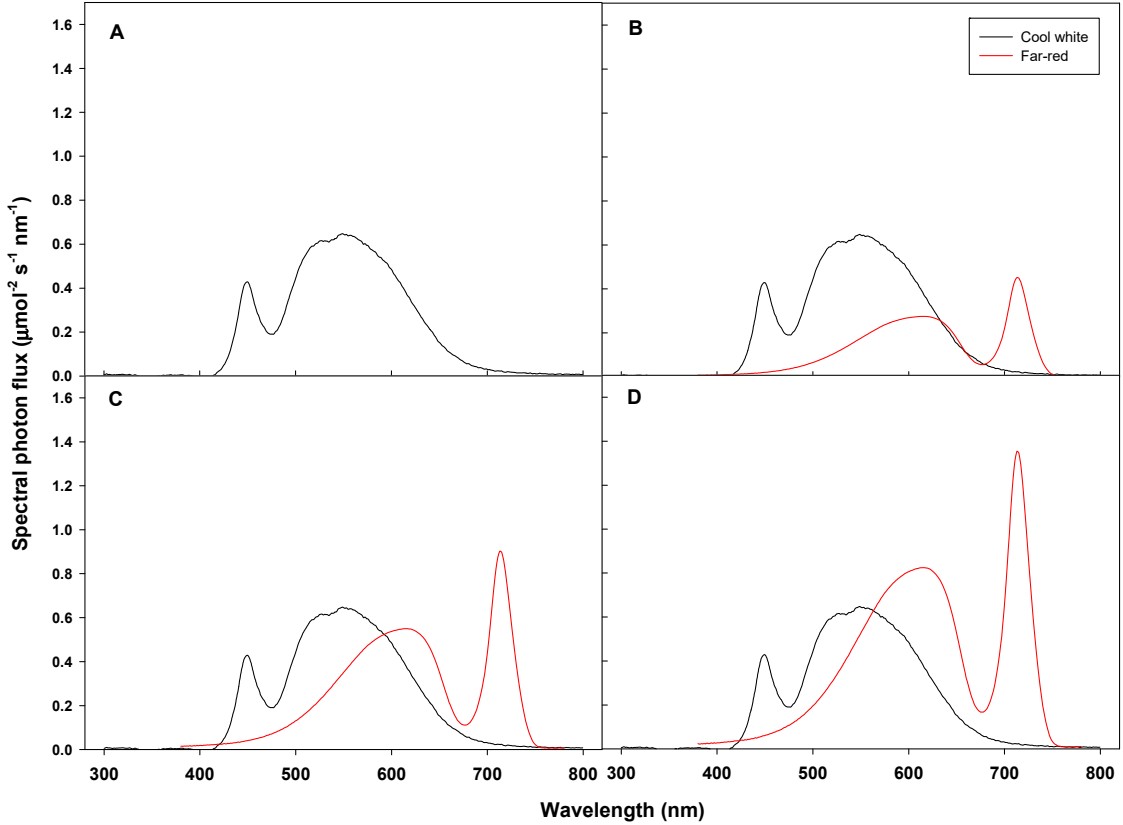

**Figure 1.** Spectral distribution of the light treatments W5F0 (**A**), W5F1 (**B**), W5F2 (**C**) and W5F3 (**D**).

*2.3. Plant Growth Analysis*

The experiment consisted of 4 treatments replicated 3 times with 8 plants per replication. Hypocotyl length, stem diameter, shoot fresh weight and dry weight of tomato and red pepper seedlings were

measured at 14 days after sowing (DAS), cucumber and gourd seedlings at 7 DAS and watermelon and bottle gourd at 9 DAS. Stem diameter was measured using a Vernier caliper (CD-20CPX, Mitutoyo Co., Kawasaki, Japan). Fresh weight was measured using a precision scale (Fx-300i, A&D Weighing, CA, USA) and dry weight after drying at 80 °C for 3 days (HB-502M, Hanbaek Co., Ltd., Bucheon, Korea). Compactness was calculated as shoot dry weight/hypocotyl length.

### 2.4. Statistical Analysis

The experimental data were analyzed by the Statistical Analysis System (SAS) for Windows version 9.4 (SAS Institute Inc., Cary, NC, USA) using Duncan's multiple range test at $\alpha < 0.05$ to compare treatments.

## 3. Results

### 3.1. Plant Growth

The effect of supplemental far-red treatments on the growth of the vegetable seedlings is shown in Figure 2, showing the effect of the different treatments on seedling growth of each species. The hypocotyl length of tomato seedlings under W5F1 was greater than or equal to W5F0 (Table 2). However, as far-red light intensity increased, it decreased even lower than the W5F0 in tomato seedllings. Although red pepper seedlings similarly showed the longest hypocotyl length in W5F1, hypocotyl length in W5F2 and W5F3 was not lower than W5F0. For cucumber, gourd, watermelon and bottle gourd seedlings, every hypocotyl length by the supplemental far-red lights was higher than W5F0 and showed the greatest features in W5F2.

The stem diameter of tomato and red pepper seedlings with W5F1 was generally greater than W5F0. The stem diameter of cucumber was the greatest with W5F2, while gourd had a greater stem diameter under all treatments than W5F0, but treatments did not differ significantly. Watermelon stem diameter with supplemental far-red light increased, while bottle gourd did not show significant differences from W5F0.

The shoot dry weights of all tomato and red pepper seedlings showed similar changes to the hypocotyl length with supplemental far-red light. The dry weights of "B-blocking" tomato seedlings were greatest under W5F1, while dry weights with W5F2 and W5F3 were significantly lower than W5F0. For cucumber and watermelon, the dry weight was the greatest in W5F2 and W5F3, respectively. The shoot dry weight of gourd was not affected by far-red light, and bottle gourd in under W5F3 was greater than in W5F0.

Supplemental far-red-enriched lights influenced the compactness of seedlings differently depending on plant species (Figure 3). Compactness is the ratio of the shoot dry weight and hypocotyl length, and higher compactness indicates the seedlings were short and denser [33]. The compactness of the "Dotaerangdia" tomato under W5F1 and W5F3 was higher than W5F0. In contrast, the compactness of the "B-blocking" tomato was the greatest under W5F1. However, the compactness of red pepper seedlings showed a clear declining trend as far-red-enriched light intensity increased. The compactness of cucumber and gourd was decreased significantly by the supplemental far-red-enriched lights, regardless of the light intensity. The compactness of watermelon under W5F1 and W5F3 was higher than W5F0, while that of bottle gourd was decreased as far-red-enriched light intensity increased.

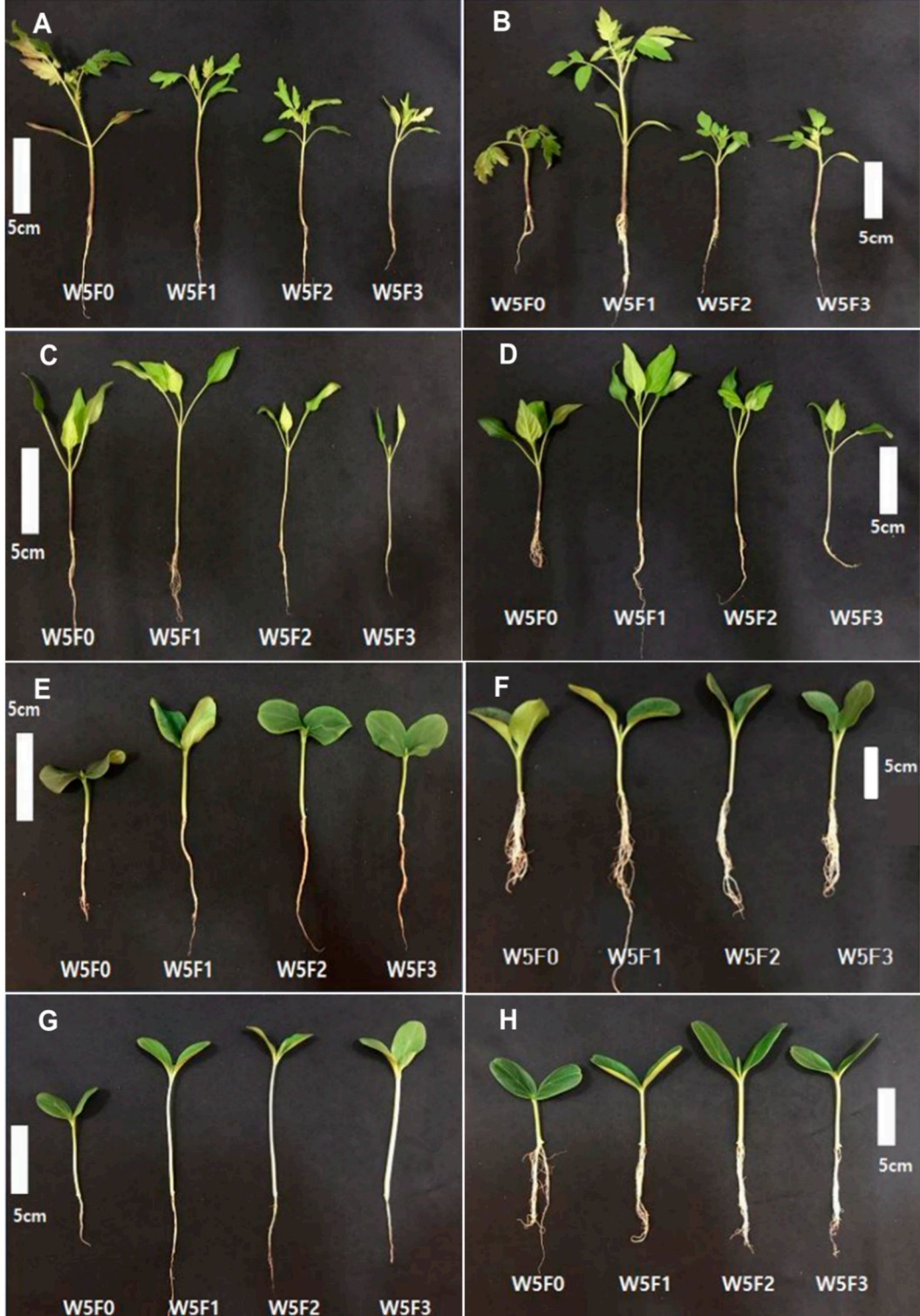

**Figure 2.** Seedlings of tomato "Dotaerangdia" (**A**) and "B-blocking" (**B**) at 14 days after sowing (DAS), red pepper "Shinhong" (**C**) and "Tantan" (**D**) at 14 DAS, cucumber (**E**) and gourd (**F**) at 7 DAS and watermelon (**G**) and bottle gourd (**H**) at 9 DAS under the different supplemental far-red-enriched LED treatments. The treatments were cool white LED:far-red LED at ratios of 5:0, 5:1, 5:2 and 5:3, or W5F0, W5F1, W5F2 and W5F3, respectively.

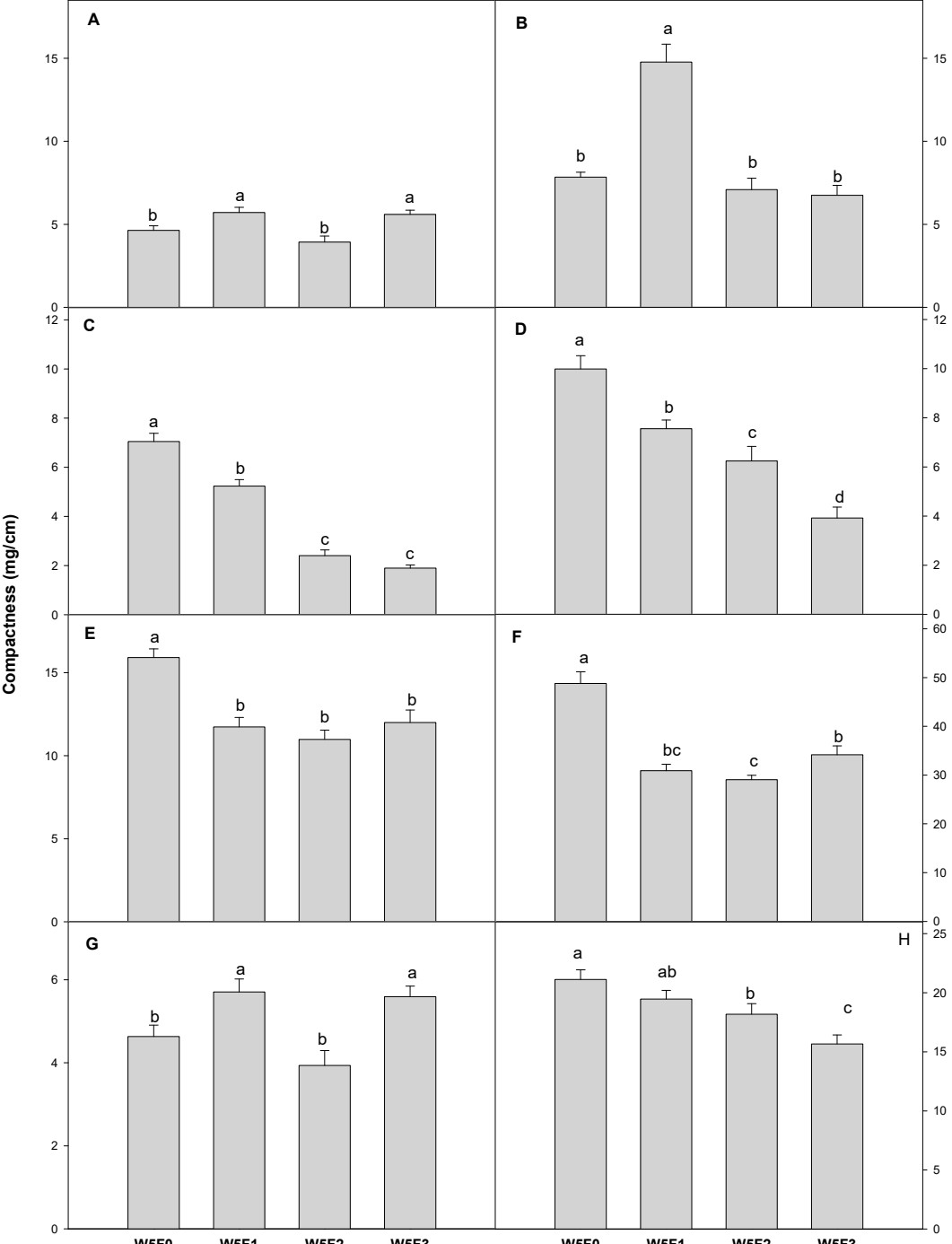

**Figure 3.** Compactness of tomato "Dotaerangdia" (**A**) and "B-blocking" (**B**) at 14 days after sowing (DAS), red pepper "Shinhong" (**C**) and "Tantan" (**D**) at 14 DAS, cucumber (**E**) and gourd (**F**) at 7 DAS and watermelon (**G**) and bottle gourd (**H**) at 9 DAS by the different supplemental far-red-enriched LED lighting. Data represent the mean of three replications with eight plants per replication per species. Vertical bars represent standard errors of means. Mean separation among treatments within each cultivar by Duncan's multiple range test at $\alpha < 0.05$.

**Table 2.** Growth of tomato at 14 days after sowing (DAS), red pepper at 14 DAS, cucumber and gourd at 7 DAS and watermelon and bottle gourd at 9 DAS by the different supplemental far-red-LED-enriched lighting. Data represent the mean of three replications with eight plants per replication per species.

| Plant | Cultivar | Treatment | Hypocotyl Length (cm) | | Stem Diameter (mm) | | Shoot | | | |
|---|---|---|---|---|---|---|---|---|---|---|
| | | | | | | | Fresh Weight (g) | | Dry Weight (g) | |
| Tomato | Dotaerangdia | W5F0 | 5.09 | b [z] | 16.1 | a | 0.947 | a | 0.049 | a |
| | | W5F1 | 6.22 | a | 16.7 | a | 0.678 | b | 0.044 | a |
| | | W5F2 | 4.94 | b | 10.3 | b | 0.245 | c | 0.015 | b |
| | | W5F3 | 4.97 | b | 11.3 | b | 0.210 | c | 0.012 | b |
| | B-blocking | W5F0 | 5.74 | a | 17.7 | b | 0.725 | b | 0.043 | b |
| | | W5F1 | 5.59 | a | 22.1 | a | 1.250 | a | 0.084 | a |
| | | W5F2 | 4.70 | b | 15.3 | c | 0.544 | c | 0.033 | bc |
| | | W5F3 | 4.68 | b | 15.3 | c | 0.478 | c | 0.031 | c |
| Pepper | Shinhong | W5F0 | 3.88 | c | 7.9 | b | 0.339 | b | 0.027 | b |
| | | W5F1 | 5.61 | a | 10.4 | a | 0.398 | a | 0.031 | a |
| | | W5F2 | 4.86 | ab | 7.1 | b | 0.156 | c | 0.011 | c |
| | | W5F3 | 4.28 | bc | 5.9 | c | 0.098 | d | 0.008 | d |
| | Tantan | W5F0 | 3.53 | d | 9.1 | b | 0.389 | b | 0.035 | b |
| | | W5F1 | 5.80 | a | 11.5 | a | 0.582 | a | 0.043 | a |
| | | W5F2 | 5.53 | b | 11.0 | a | 0.402 | b | 0.034 | b |
| | | W5F3 | 4.83 | c | 9.3 | b | 0.261 | c | 0.018 | c |
| Cucumber | Joeunbaegdadagi | W5F0 | 2.27 | c | 16.7 | b | 0.402 | c | 0.035 | c |
| | | W5F1 | 3.53 | b | 18.5 | b | 0.501 | b | 0.041 | b |
| | | W5F2 | 4.59 | a | 23.0 | a | 0.598 | a | 0.049 | a |
| | | W5F3 | 3.74 | b | 19.1 | b | 0.494 | b | 0.044 | b |
| Gourd | Heukjong | W5F0 | 2.93 | b | 26.7 | a | 1.742 | a | 0.014 | a |
| | | W5F1 | 4.75 | a | 29.0 | a | 1.910 | a | 0.014 | a |
| | | W5F2 | 4.91 | a | 28.0 | a | 1.743 | a | 0.014 | a |
| | | W5F3 | 4.57 | a | 28.6 | a | 1.875 | a | 0.015 | a |
| Watermelon | Sambokkul | W5F0 | 4.51 | d | 19.7 | b | 0.447 | c | 0.020 | c |
| | | W5F1 | 7.53 | c | 25.2 | a | 0.876 | b | 0.042 | a |
| | | W5F2 | 9.31 | a | 24.5 | a | 0.869 | b | 0.036 | b |
| | | W5F3 | 8.13 | b | 27.5 | a | 1.016 | a | 0.045 | a |
| Bottle gourd | Bulrojangsaeng | W5F0 | 3.71 | c | 30.9 | ab | 1.167 | c | 0.077 | b |
| | | W5F1 | 4.27 | b | 28.3 | b | 1.265 | bc | 0.083 | b |
| | | W5F2 | 5.40 | a | 35.9 | a | 1.565 | a | 0.097 | a |
| | | W5F3 | 5.14 | a | 31.3 | ab | 1.374 | b | 0.079 | b |

[z] Mean separation among treatments within columns and within each cultivar by Duncan's multiple range test at $\alpha < 0.05$.

### 3.2. Plant Growth Analysis for Phytochrome Stationary State (PSS)

The hypocotyl length of each plant responded differently to PSS values and generally resulted in a quadratic regression and a peak value (Figure 4). Maximum hypocotyl lengths of tomato "Dotaerangdia" and "B-blocking" were 6.27 cm at PSS 0.73 and 6.00 cm at PSS 0.77, respectively. Maximum hypocotyl lengths of red pepper were 5.68 cm at PSS 0.71 for "Shinhong" and 5.58 cm at PSS 0.69 for "Tantan". In the case of cucumber, maximum hypocotyl length was 4.06 cm at PSS 0.61; for gourd, it was 4.86 cm at PSS 0.66. The maximum hypocotyl length of watermelon was 8.59 cm at PSS 0.60, while bottle gourd did not show a significant quadratic regression.

The dry weight of each plant responded differently to PSS values (Figure 5). Although there were highly significant quadratic relationships for tomato and red pepper seedlings, cucumber, gourd, watermelon and bottle gourd seedlings did not show clear relationships of dry weight to PSS values. The maximum dry weight of the "Dotaerangdia" and "B-blocking" tomato was 0.066 g at PSS 0.77 and 0.100 g at PSS 0.74, respectively. For red pepper, the maximum dry weight was 0.046 g at PSS 0.77 for "Shinhong" and 0.052 g at PSS 0.75 for "Tantan".

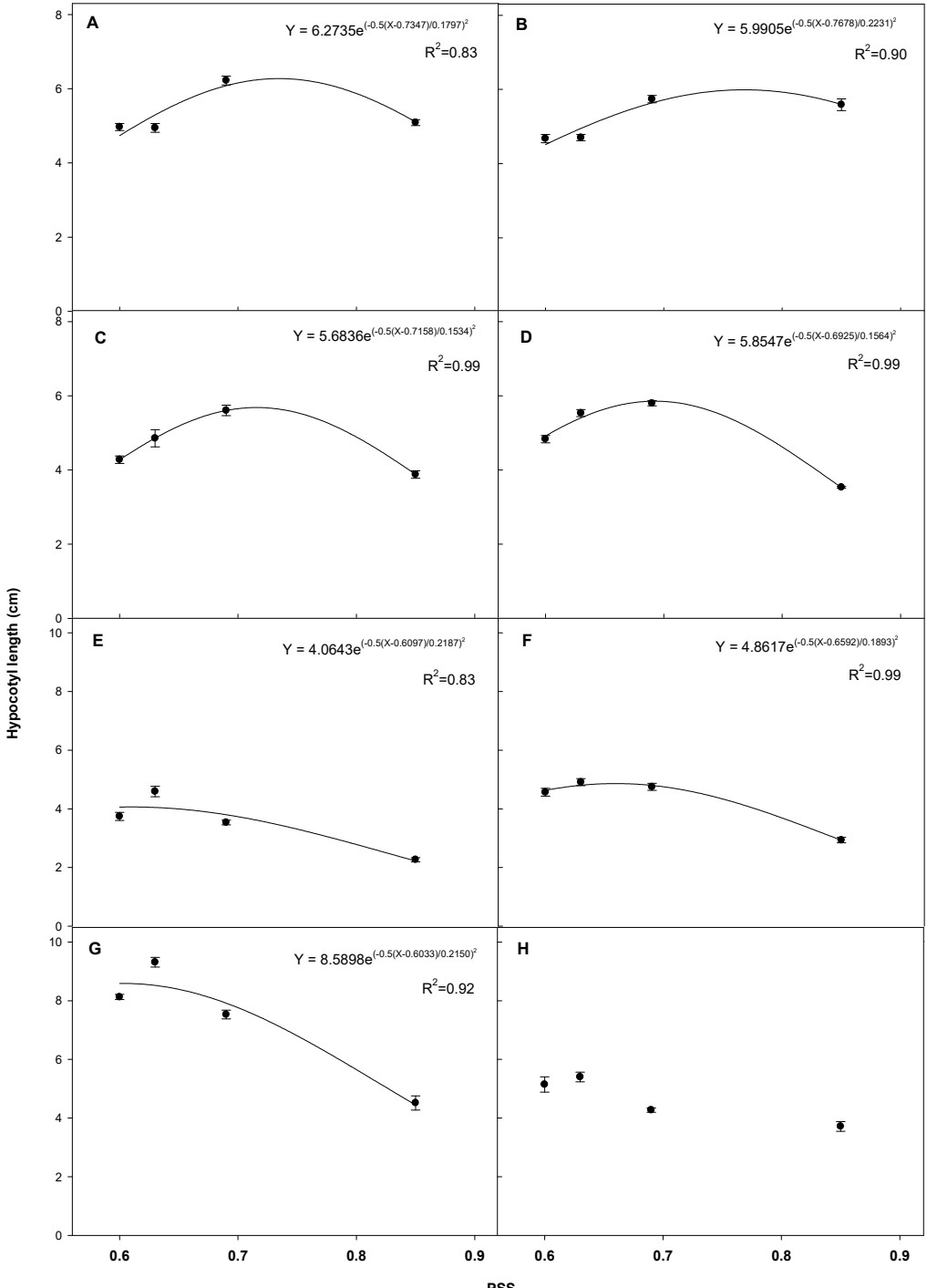

**Figure 4.** Regression of hypocotyl length with phytochrome stationary state (PSS). The value of each point is the average hypocotyl length of "Dotaerangdia" (**A**) and "B-blocking" (**B**) tomato at 14 days after sowing (DAS), "Shinhong" (**C**) and "Tantan" (**D**) red pepper at 14 DAS, cucumber (**E**) and gourd (**F**) at 7 DAS and watermelon (**G**) and bottle gourd (**H**) at 9 DAS by the different supplemental far-red-enriched LED lighting treatments. Data represent the mean of three replications with eight plants per replication per species/cultivar. Vertical bars represent standard errors of means.

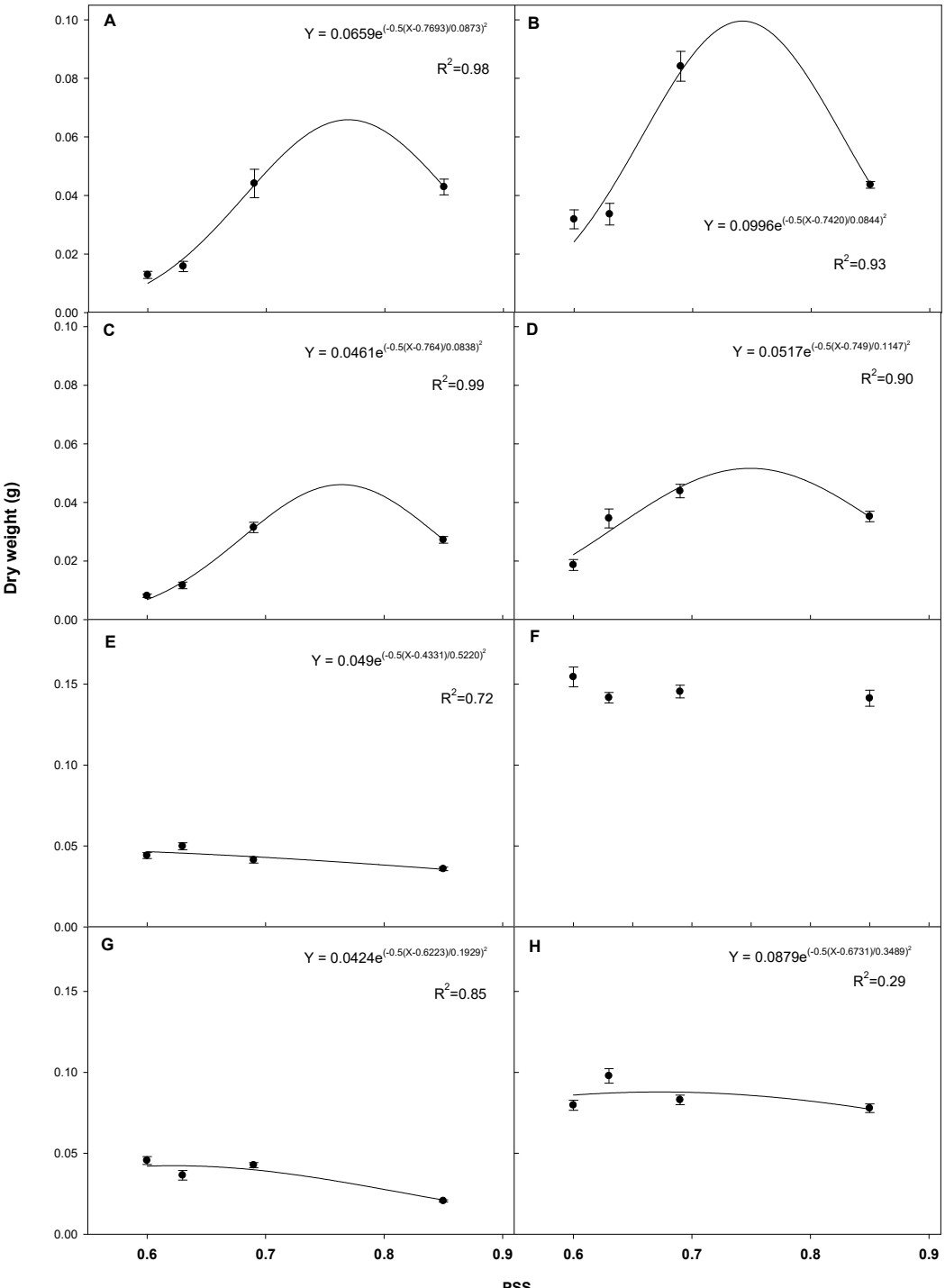

**Figure 5.** Regression of shoot dry weight versus phytochrome stationary state (PSS). The value of each point is the average dry weight of the "Dotaerangdia" (**A**) and "B-blocking" (**B**) tomato at 14 days after sowing (DAS), "Shinhong" (**C**) and "Tantan" (**D**) red pepper at 14 DAS, cucumber (**E**) and gourd (**F**) at 7 DAS and watermelon (**G**) and bottle gourd (**H**) at 9 DAS under the different supplemental far-red-enriched LED lighting treatments. Data represent the mean of three replications with eight plants per replication per species/cultivar. Vertical bars represent standard errors of means.

## 4. Discussion

The hypocotyl length of each species, and cultivar within species in some instances, responded differently to PSS values and resulted in a peak for all species except bottle gourd seedlings. A similar result of quadratic relationships with peak values was reported with *Crepidiastrum denticulatum* [34], similar to our results with tomato and red pepper seedlings. The PSS value for maximum hypocotyl length by supplemental far-red light was 0.69–0.77 for tomato and red pepper seedlings (Figure 3). The maximum hypocotyl length of "Dotaerangdia" and "B-blocking" tomato was 42.9 cm at PSS 0.73 and 4.4% greater at PSS 0.77 than the control at PSS 0.85, respectively. Those of "Shinhong" and "Tantan" red pepper were 46.5 at PSS 0.71 and 58.2% greater at PSS 0.69 than the control, respectively. However, the peak values for cucumber, gourd and watermelon were not clearly different from controls in contrast to those of tomato and red pepper seedlings.

Shoot dry weight varied with supplemental far-red light spectral intensity and quality by plant species, showing clear peak values for tomato and red pepper seedlings. However, cucumber, gourd, watermelon and bottle gourd seedlings did not show clear relationships between their dry weights and PSS values. Usually, the cucumber and watermelon (Cucurbitaceae scion) and gourd and bottle gourd (Cucurbitaceae rootstock) were grafted before the true leaves occurred, and the Solanaceae seedlings were grafted when three to four leaves occurred. It is the reason why the far-red light response in the Solanaceae was much clearer. The PSS value at maximum shoot dry weight ranged from 0.74 to 0.77 for tomato and red pepper seedlings (Figure 4). The shoot dry weight increase of "Dotaerangdia" and "B-blocking" tomato was 53.6% and 128.4% of the control value, when their dry weight was at PSS 0.73 and 0.77, respectively. Those of "Shinhong" and "Tantan" red pepper were 69.1% and 46.6% compared to the control, when the PSS values were at 0.74 and 0.75, respectively. Stem diameter in the far-red treatments varied with species/cultivars, while no clear trends were observed (Table 1). In this study, the compactness of each species/cultivar showed different responses to supplemental far-red lighting. A decrease in leaf thickness is known to be one factor decreasing compactness, and this may have affected the compactness values in our study. This is validated by a previous study showing that supplemental far-red lighting resulted in thinner leaves [16].

The hypocotyl length and dry weight of the vegetable seedlings in this study increased to peak values as the far-red light intensity increased. Plant growth enhancement by supplemental far-red light is generally explained as a synergistic effect on photosynthesis [17,35], with a stimulated distribution of photosynthetic products to the shoot [18]. These growth and morphological changes at relatively low PSS values are a result of phytochrome-mediated shade avoidance responses [18] [36,37]. Previous studies showed that plant growth and morphology improved with far-red light treatments in lettuce [1,4–6], tomato [7], squash [8], red pepper [9], cucumber [38], snapdragon [10] and geranium (*Pelargonium* spp.) [29] seedlings. Kalaitzoglou et al. [39] reported that decreasing PSS from 0.88 to 0.80 increased the total dry weight and plant height of young tomato seedlings. However, the results in our study show that plant responses such as growth, including hypocotyl elongation, can vary significantly among species, from positive to negative to unaffected. On the one hand, the effect of a phytochrome-mediated far-red high irradiance response (FR-HIR) under monochromatic far-red light treatments has been shown to improve the growth and morphology of seedlings generally [20,40,41]. On the other hand, hypocotyl elongation has been reported to be inhibited by far-red light treatment in some species such as radish (*Raphanus sativus* L.) [42]. Schäfer et al. found that the peaks in far-red light action spectra and intensity response curves for the inhibition of hypocotyl growth in white mustard seedlings were 740 nm and 21 $\mu mol \cdot m^{-2} \cdot s^{-1}$, respectively [43,44].

With supplemental far-red light treatments, the far-red light effect can be detected by using R:FR and PSS in general. In this study, as far-red light intensity increased by supplemental far-red light treatments, blue light (400–500 nm), photosynthetic photon flux (PPF) and total photon flux (TPF) also increased (Table 2). According to Hogewoning et al. [45], growth and leaf responses were not affected by an increase in blue light intensity, up to 22 $\mu mol \cdot m^{-2} \cdot s^{-1}$. Similarly, Fan et al. [46] reported that tomato seedlings showed no substantial gain in growth from additional PPF above

300 μmol·m$^{-2}$·s$^{-1}$. Based on this information, we suggest that vegetable seedling growth in the far-red light treatments in this study would not be affected by an increase in blue light (ranging from 42.4 to 53.2 μmol·m$^{-2}$·s$^{-1}$). However, previous research suggested that some phytochrome responses were dose-dependent, and their effects on growth and morphology decreased above a threshold of light intensity [34]. Plant responses to far-red light have been shown to be a function of the R:FR ratio, and light intensity can also regulate the shade avoidance response. In general, increasing light intensity decreases the magnitude of plant responses to far-red light effects. For example, stem elongation of sunflower (*Helianthus* spp.) increased with a reduction in the R:FR ratio (from 4.52 to 0.85) under both a low light intensity (157 μmol·m$^{-2}$·s$^{-1}$) and a high intensity (421 μmol·m$^{-2}$·s$^{-1}$), but the response was reduced under much higher light intensity [47]. Similarly, R:FR signaling through phytochrome B promoted branching of *Arabidopsis thaliana* under low light intensity (160 μmol·m$^{-2}$·s$^{-1}$), but the effects diminished under higher light intensity (280 μmol·m$^{-2}$·s$^{-1}$), indicating that high light intensity at least partially negates the effects of phytochrome-mediated signaling in plants [48]. Therefore, the effects of supplemental far-red-enriched lighting on growth including hypocotyl length were affected by the increase in PPF and TPF. An appropriate range of far-red light intensity could modulate plant growth and morphology for specific purposes. For example, it would be advantageous for grafting materials, such as scion and rootstock of seedlings, controlling the stem length of cut flowers [47], and training of many vegetable crops in protected horticulture [49].

## 5. Conclusions

The results of this study show that far-red-enriched lighting affects plant growth and morphology, but responses can greatly vary depending on light intensity and plant species. Growth, including hypocotyl length of tomato and red pepper seedlings, reached peak increases at certain PSS values and decreased with PSS values above that. However, there was no clear relationship between PSS and growth or hypocotyl length for cucumber or watermelon. Therefore, careful consideration must be given in the selection of appropriate far-red light intensities for each plant species and is necessary for the application of supplemental far-red-enriched lighting with LEDs to achieve a desired plant growth and morphology.

**Author Contributions:** Conceptualization, methodology, data curation, formal analysis, investigation, writing, H.H.; conceptualization, methodology, writing, project administration, S.A.; methodology, data curation, formal analysis, B.L.; conceptualization, supervision, validation, writing, review and editing, funding acquisition, C.C. All authors have read and agreed to the published version of the manuscript.

**Funding:** This research received no external funding.

**Acknowledgments:** This research was funded by the Rural Development Administration (PJ013840) "Development of closed system type seedling production system to produce standard fruit vegetable seedlings linked with a grafting robot".

**Conflicts of Interest:** The authors declare no conflict of interest.

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
