# Peer review of "Improvement of Growth and Morphology of Vegetable Seedlings with Supplemental Far-Red Enriched LED Lights in a Plant Factory"

_horticulturae, doi:10.3390/horticulturae6040109_

Round 1

Reviewer 1 Report

The article entitled ‘Improvement of growth and morphology of fruit vegetable seedlings with supplemental far-red enriched LED lights in a plant factory’ describes experiment with the use of cool light LED with addition of different intensities of far-red light in growth chamber in order to improve growth and morphological parameters of several vegetable seedlings. The issue raised in this manuscript may have practical implications for the development of specific recommendations for the use of far red light in the cultivation of various vegetable species.

I have several remarks, which in my opinion should be improved in the manuscript before publication:

L. 37 – ‘…plant growth and morphology increased…’ – morphology increased is not adequate statement. Please, expand and specify what is morphology increase.

L. 41 – similarly as above, please specify what is improving photomorphogenesis

L. 45 – shade avoidance response appears first in the text, please add abbreviation (SAR), which is used further in the text.

L. 66-110 – Please, add more specific description of plant cultivation manner and sampling for analyses: how many plants of each species were cultivated within one treatment (is it 128 plants, as 128-cell plug trays were used?), how many plants were taken for each analysis, how many measurements were taken for statistical analysis?

L. 101-102 – Description of hypocotyl length analyses are described for only 4 species, when experiment consisted of 6 species. Please, add gourd and bottle gourd to the description.

References: please, check if all formatting is proper. Also, only approx.. 20% of referred literature has been published within last 5 years. Citing more current literature would be with benefit for the relevance of the article.

Also, I encourage authors to improve used language by English editing.  

Author Response

Response to Reviewer 1 Comments

Thank you for your concise comments. We understand your concern about several points.

The answers to the comments are organized as follows:

Point 1: Line 37, ‘…plant growth and morphology increased…’ – morphology increased is not adequate statement. Please, expand and specify what is morphology increase.

Response 1: We changed the verb from “increased” to “improved”. (Line 37)

Point 2: Line 41, similarly as above, please specify what is improving photomorphogenesis

Response 2: We changed the phrase from “improving photomorphogenesis” to “improving photomorphogenesis including stem development”. (Line 41)

Point 3: Lines 45, shade avoidance response appears first in the text, please add abbreviation (SAR), which is used further in the text.

Response 3:  We added abbreviation term, shade avoidance response (SAR) (Line 46-47)

Point 4: Line 66-110: Please, add more specific description of plant cultivation manner and sampling for analyses: how many plants of each species were cultivated within one treatment (is it 128 plants, as 128-cell plug trays were used?), how many plants were taken for each analysis, how many measurements were taken for statistical analysis

Response 4: We added sampling method and added description on each figure and table.

Point 5: Lines 101-102: Description of hypocotyl length analyses are described for only 4 species, when experiment consisted of 6 species. Please, add gourd and bottle gourd to the description.

Response 5: We added the description about gourd and bottle gourd. (Line 109)

Point 6: Refereces: please, check if all formatting is proper. Also, only approx.. 20% of referred literature has been published within last 5 years. Citing more current literature would be with benefit for the relevance of the article.

Response 6: We added more recent reference and carefully revised the references.

Reviewer 2 Report

“Improvement of growth and morphology of fruit vegetable seedlings with supplemental far-red enriched LED lights in a plant factory”

The investigation is interesting, giving the new insight how light quality control can improve the cultivation of vegetable. The manuscript provides information for growers on selection of appropriate light intensity for different vegetable species and cultivars to achieve desired plant growth and morphology. The hypothesis however, was not clearly stated and it should be added. The manuscript is aimed to investigate the effect of supplemental far-red light intensities to control growth of young vegetable seedlings. Nevertheless, in their conclusion, the authors stated plants responded differently and it depends on light intensity and plant species. To my opinion, the conclusion should be written more precisely.

Also, the authors should consider using some help for English language editing of style and grammar since some sentences are incomprehensible and it is hard to follow. Parts of the discussion is merely repeating of the results and in-depth discussion of the results is lacking.

In M&M section there should be methodology explained in more details. E.g. how were the seedling grown? In soil, hydroponically or in nutrient solution? The authors only wrote that they watered seedlings every 2-3 days with nutrient solution. What exactly ratios of W and F lights describe? With different number of lights or the intensity can be regulated? It should be defined more precisely.

The authors calculated compactness of their seedlings. However, they didn`t mention in any part of the manuscript why is it important and its significance for the growers. It should be explained more.

The light intensity (PPF and FTP) changed between treatments for about 50 μmol m-2 s-1 but this change was not due only the increase in far-red light intensity but also from different ranges of visible light. Although the PPF and FTP didn`t show considerable differences (except in WF1 treatment), to my opinion, the authors should not refer that changes in seedlings morphology came from far-red light intensity.

Figures 4H and 5F should be corrected, the trend lines, Y and R2 are missing.

However, my main concern about this investigation is the fact that the methodology includes only few morphological parameter. To my understanding, even the PSS value is estimated according to well-known values (if I am mistaking, it should be explained in more details how the authors measured it and abbreviations from the formula should be explained and described).

Best regards.

Author Response

Response to Reviewer 2 Comments

Point 1: In M&M section there should be methodology explained in more details. E.g. how were the seedling grown? In soil, hydroponically or in nutrient solution? The authors only wrote that they watered seedlings every 2-3 days with nutrient solution. What exactly ratios of W and F lights describe? With different number of lights or the intensity can be regulated? It should be defined more precisely.

 Response 1: We added more detail methodology in materials and methods part. (Line 73-74, 81-82, 83-85)

Point 2: The authors calculated compactness of their seedlings. However, they didn`t mention in any part of the manuscript why is it important and its significance for the growers. It should be explained more.

Response 2: We added the description about the importance of seedlings compactness and attached related reference. (Line 151-152)

Point 3: The light intensity (PPF and FTP) changed between treatments for about 50 μmol m-2 s-1 but this change was not due only the increase in far-red light intensity but also from different ranges of visible light. Although the PPF and FTP didn`t show considerable differences (except in WF1 treatment), to my opinion, the authors should not refer that changes in seedlings morphology came from far-red light intensity.

 Response 3: We add a review of the other wavelengths. In addition to the FR effect, we added a review of the PPF and TPF that have changed together. (Line 238-255)

Point 4: Figures 4H and 5F should be corrected, the trend lines, Y and R2 are missing.

Response 4: PSS effects on hypocotyl length of bottle gourd and those on dry weight of gourd did not showed regression (Gaussian regression model) by SAS PROC REG. (Figure 4 and 5)

Point 5: However, my main concern about this investigation is the fact that the methodology includes only few morphological parameter. To my understanding, even the PSS value is estimated according to well-known values (if I am mistaking, it should be explained in more details how the authors measured it and abbreviations from the formula should be explained and described).

Response 5: We added more detail description about the PSS value. (Line 92-98)

Reviewer 3 Report

Review

The reviewed article presents the results of research on a very interesting issue related to the use of additional far red LED lights in plant factory.

Introduction

Short and concise, containing basic information related to the issue under consideration.

  1. 2 line 60. Fragment ... and release low temperature with long time span than the others ..... requires supplementing, as there is no context related to it in the introduction, this fragment is unclear.

Materials and Methods.

  1. 2 line 73. There is no information on which seed sample the research was performed. Information ....... were sown on a 128 cell plug tray ....... is not enough, because it may mean that it was 128 cell plug tray x 8 species x 4 light treatment = 4096 seedlings - which is a very large experimental sample, but it can also mean 128/8/4 = 4 - which is, a very small experimental sample, with a very large error resulting, among others, from from the quality of the seeds. The difference is significant and this information was not found in the publication. There is also no information about the quality of the seeds, the factor that could have influenced on the obtained results. There was also no information on the medium on which the seeds were sown, the method of sowing, the method of filling the cells in the tray, the density of the substrate in a single cell g/cm3 and the variability of this density in the experiment, which in the case of e.g. germination time will have a significant impact, but also the availability of water and nutrients. The lack of this information makes it difficult to assess the results obtained.
  2. 2 line 79. There is no detailed information on how the lamps were installed, on what height, in what position the lamps were in relation to each other (T5 vs HT251), and how the lighting parameters reaching the plants were controlled (illuminance ?). No information was provided on the source of the spectral distribution in Fig. 1. Was the measurement made or the manufacturer's characteristics used?

2.3. Plant growth analysis.

  1. 4 line 101. Only one of the measuring instruments is given and the others are omitted, please complete with information about the accuracy of the measurements made
  2. 4 line 104. Compactness is a parameter commonly used to define the mass to volume ratio and in the literature it generally refers to the bulk density of soil or substrate. Wouldn't it be better to name this indicator differently?

Results

In Table 2, apart from the average, the parameters indicating the variability of the parameter should also be given.

Discussion

  1. 10 line 82 and 191. The first two paragraphs of the discussion describe the results. It seems too detailed.

Conclusions

Can the PSS range be given in the conclusions (as in the abstract) to obtain the maximum hypocotyl length in tomato and red pepper seedling?

References

The list of references should be standardized, especially with regard to the use of upper and lower case letters.

Author Response

Response to Reviewer 3 Comments

Point 1: line 60-61, Fragment ... and release low temperature with long time span than the others ..... requires supplementing, as there is no context related to it in the introduction, this fragment is unclear.

Response 1: We removed the words “and release low temperature with long time span than the others”. (Line 61)

Point 2: line 73-74, There is no information on which seed sample the research was performed. Information ....... were sown on a 128 cell plug tray ....... is not enough, because it may mean that it was 128 cell plug tray x 8 species x 4 light treatment = 4096 seedlings - which is a very large experimental sample, but it can also mean 128/8/4 = 4 - which is, a very small experimental sample, with a very large error resulting, among others, from the quality of the seeds. The difference is significant and this information was not found in the publication. There is also no information about the quality of the seeds, the factor that could have influenced on the obtained results. There was also no information on the medium on which the seeds were sown, the method of sowing, the method of filling the cells in the tray, the density of the substrate in a single cell g/cm3 and the variability of this density in the experiment, which in the case of e.g. germination time will have a significant impact, but also the availability of water and nutrients. The lack of this information makes it difficult to assess the results obtained.

Response 2: We added more detail information about the plant materials and cultivations. (Line 73-74, 81-82)

Point 3: line 79-80, There is no detailed information on how the lamps were installed, on what height, in what position the lamps were in relation to each other (T5 vs HT251), and how the lighting parameters reaching the plants were controlled (illuminance ?). No information was provided on the source of the spectral distribution in Fig. 1. Was the measurement made or the manufacturer's characteristics used?

Response 3: We added more detail information about the light treatment. (Line 77-78, 83-85)

Point 4: line 101, Only one of the measuring instruments is given and the others are omitted, please complete with information about the accuracy of the measurements made

Response 4: We added the information of measuring instruments. (Line 111-112)

Point 5: line 104, Compactness is a parameter commonly used to define the mass to volume ratio and in the literature it generally refers to the bulk density of soil or substrate. Wouldn't it be better to name this indicator differently?

 Response 5: We added the description of compactness and attached related reference. (Line 151-152)

Point 6: Table 2, apart from the average, the parameters indicating the variability of the parameter should also be given.

Response 6: We modified table 2, and added description about the treatment. (Line 147-148)

Point 7: line 82-91: The first two paragraphs of the discussion describe the results. It seems too detailed.

Response 7: Thank you for your concise comments. We understand your concern about the discussion parts.

Point 8: Conclusions, Can the PSS range be given in the conclusions (as in the abstract) to obtain the maximum hypocotyl length in tomato and red pepper seedling?

Response 8: While more data points will make the analyses more robust, we believe the regression curve is still trustworthy with few data points if the model fits the data with high coefficient of determination. In this study, we applied Gaussian regression model, one of the most common type of models for biological responses. The regression analyses returned fitting curves with very high coefficients of determination in Fig. 4A, B, C and D (R2 ranged from 0.90 to 0.99), which means these 4 data points had a strong impact on the fitting curve. Even if more data points were to be added, the fitting curves would not drastically change unless R2 became very low. Therefore, we think that these PSS range are believable.

Point 9: References, The list of references should be standardized, especially with regard to the use of upper and lower case letters.

Response 9: We carefully revised the references.

Round 2

Reviewer 1 Report

Dear Authors,

Thank you for response to my remarks and all improvements made to the text.

In my opinion manuscript is suitable for publication.

Author Response

Response to Academic Editor

Thank you for your concise comments. We understand your concern about several points.

The answers to the comments are organized as follows:

Point 1: line 59-60, Please add how they are related, or as PSS increases what happens to shoot elongation.

Response 1: We added sentence “PSS value from 0.70 to 0.85 has shown a negative linear correlation between shoot lengths of the plant”

Point 2: line 79, Cite source of information or commercial product

Response 2: We added a reference to the nutrient solution.

Point 3: line 84, Is intensity the correct word? Do you mean numbers of each type? If so, does 5:1 mean 10 cool white and 2 far-red?

Response 3: W5F1 means 10 cool white and 2 far-red.

Point 4: line 93-94, Using what to collect this?

Response 4: We acquired 12 points data to get the average of the cultivation space using the tray.

Point 5: line 107-113, Is this correct? Initally was awkward. Is this shoot only or whole plant? The last sentence says shoot. So roots were excised?

Response 5: In this study, we investigated with data from the shoot only.

Point 6: line 124-128, Not in Doteaerangdia

Response 6: We revised it.

Point 7: line 247, Citation number, not year.

Response 7: We changed the citation format.

Point 8: line 257, Does this mean higher than 421 μmol∙m-2∙s-1?

Response 8: This indicates that 421 μmol∙m-2∙s-1 is relatively high light intensity.

Reviewer 2 Report

Most of my comments were answered and appropriate additions were made

Author Response

Response to Academic Editor

Thank you for your concise comments. We understand your concern about several points.

The answers to the comments are organized as follows:

Point 1: line 59-60, Please add how they are related, or as PSS increases what happens to shoot elongation.

Response 1: We added the sentence “PSS value from 0.70 to 0.85 has shown a negative linear correlation between shoot lengths of the plant”

Point 2: line 79, Cite source of information or commercial product

Response 2: We added a reference to the nutrient solution.

Point 3: line 84, Is intensity the correct word? Do you mean numbers of each type? If so, does 5:1 mean 10 cool white and 2 far-red?

Response 3: W5F1 means 10 cool white and 2 far-red.

Point 4: line 93-94, Using what to collect this?

Response 4: We acquired 12 points data to get the average of the cultivation space using the tray.

Point 5: line 107-113, Is this correct? Initally was awkward. Is this shoot only or whole plant? The last sentence says shoot. So roots were excised?

Response 5: In this study, we investigated with data from the shoot only.

Point 6: line 124-128, Not in Doteaerangdia

Response 6: We revised it.

Point 7: line 247, Citation number, not year.

Response 7: We changed the citation format.

Point 8: line 257, Does this mean higher than 421 μmol∙m-2∙s-1?

Response 8: This indicates that 421 μmol∙m-2∙s-1 is relatively high light intensity.

Reviewer 3 Report

In the presented revised article, my comments were answered and appropriate additions were made.

Author Response

(The authors gave the same response as above.)
